# Vacuolar Proton Pyrophosphatase Is Required for High Magnesium Tolerance in *Arabidopsis*

**DOI:** 10.3390/ijms19113617

**Published:** 2018-11-16

**Authors:** Yang Yang, Ren-Jie Tang, Baicong Mu, Ali Ferjani, Jisen Shi, Hongxia Zhang, Fugeng Zhao, Wen-Zhi Lan, Sheng Luan

**Affiliations:** 1Nanjing University-Nanjing Forestry University Joint Institute for Plant Molecular Biology, College of Life Sciences, Nanjing University, Nanjing 210093, China; yangyang_1604@163.com (Y.Y.); mubaicong@smail.nju.edu.cn (B.M.); fgzhao@nju.edu.cn (F.Z.); 2Department of Plant and Microbial Biology, University of California, Berkeley, CA 94720, USA; rjtang@berkeley.edu; 3Department of Biology, Tokyo Gakugei University, Koganei, Tokyo 184-8501, Japan; ferjani@u-gakugei.ac.jp; 4Nanjing University-Nanjing Forestry University Joint Institute for Plant Molecular Biology, Key Laboratory of Forest Genetics and Biotechnology, Nanjing Forestry University, Nanjing 210037, China; jshi@njfu.edu.cn; 5College of Agriculture, Ludong University, Yantai 264025, China; hxzhang@sibs.ac.cn

**Keywords:** vacuolar H^+^-pyrophosphatase, *AtAVP1*, cellular PPi homeostasis, high-Mg tolerance

## Abstract

Magnesium (Mg^2+^) is an essential nutrient in all organisms. However, high levels of Mg^2+^ in the environment are toxic to plants. In this study, we identified the vacuolar-type H^+^-pyrophosphatase, AVP1, as a critical enzyme for optimal plant growth under high-Mg conditions. The *Arabidopsis*
*avp1* mutants displayed severe growth retardation, as compared to the wild-type plants upon excessive Mg^2+^. Unexpectedly, the *avp1* mutant plants retained similar Mg content to wild-type plants under either normal or high Mg conditions, suggesting that AVP1 may not directly contribute to Mg^2+^ homeostasis in plant cells. Further analyses confirmed that the *avp1* mutant plants contained a higher pyrophosphate (PPi) content than wild type, coupled with impaired vacuolar H^+^-pyrophosphatase activity. Interestingly, expression of the *Saccharomyces cerevisiae* cytosolic inorganic pyrophosphatase1 gene *IPP1*, which facilitates PPi hydrolysis but not proton translocation into vacuole, rescued the growth defects of *avp1* mutants under high-Mg conditions. These results provide evidence that high-Mg sensitivity in *avp1* mutants possibly resulted from elevated level of cytosolic PPi. Moreover, genetic analysis indicated that mutation of *AVP1* was additive to the defects in *mgt6* and *cbl2 cbl3* mutants that are previously known to be impaired in Mg^2+^ homeostasis. Taken together, our results suggest AVP1 is required for cellular PPi homeostasis that in turn contributes to high-Mg tolerance in plant cells.

## 1. Introduction

Inorganic pyrophosphate (PPi) is an intermediate compound generated by a wide range of metabolic processes, including biosynthesis of various macromolecules such as proteins, DNA, RNA, and polysaccharides [1]. Being a high-energy phosphate compound, PPi can serve as a phosphate donor and energy source, but it can, at high levels, become inhibitory to cellular metabolism [2,3,4]. To maintain an optimal PPi level in the cytoplasm, timely degradation of excessive PPi is carried out by two major types of enzymes: soluble inorganic pyrophosphatases (sPPases) and proton-translocating membrane-bound pyrophosphatases (H^+^-PPases) [1,5,6]. The importance of maintaining an optimal cellular PPi level has been demonstrated in several different organisms. Genetic mutations that lead to the absence of sPPase activity affects cell proliferation in *Escherichia coli* [7]. In yeast, inorganic pyrophosphatase is indispensable for cell viability because loss of its function results in cell cycle arrest and autophagic cell death associated with impaired NAD^+^ depletion [8,9].

In *Arabidopsis*, a tonoplast-localized proton-pumping pyrophosphatase AVP1 was shown to be the key enzyme for cytosolic PPi metabolism in different cell types of various plants [10,11,12]. This enzyme activity has been correlated with the important function that AVP1 plays in many physiological processes [1,13,14]. Arabidopsis *fugu5* mutants lacking functional AVP1 show elevated levels of cytosolic PPi and display heterotrophic growth defects resulting from the inhibition of gluconeogenesis [13,15]. This important role in controlling PPi level in plant cells is reinforced by a recent study showing that higher-order mutants defective in both tonoplast and cytosolic pyrophosphatases display much severe phenotypes including plant dwarfism, ectopic starch accumulation, decreased cellulose and callose levels, and structural cell wall defects [16]. Moreover, the tonoplast-localized H^+^-PPase AVP1 appears to be a predominant contributor to the regulation of cellular PPi levels because the quadruple knockout mutant lacking cytosolic PPase isoforms *ppa1 ppa2 ppa4 ppa5* showed no obvious phenotypes [16]. Interestingly, in companion cells of the phloem, AVP1 was also shown to be localized to the plasma membrane [17] and function as a PPi synthase that contribute to phloem loading, photosynthate partitioning, and energy metabolism [18,19,20]. On the other hand, AVP1 is also believed to contribute to the establishment of electrochemical potential across the vacuole membrane, which is important for subsequent vacuolar secondary transport and ion sequestration [21,22]. Constitutive overexpression of AVP1 improves the growth and yield of diverse transgenic plants under various abiotic stress conditions—including drought, salinity, as well as phosphorus (P) and nitrogen (N) deficiency—although the mechanism remains to be fully understood [23,24,25,26,27]. Taken together, AVP1 serves as a multi-functional protein involved a variety of physiological processes in plants, some of which await to be fully understood.

Magnesium (Mg) is an essential macronutrient for plant growth and development, functioning in numerous biological processes and cellular functions, including chlorophyll biosynthesis and carbon fixation [28,29]. Either deficiency or excess of Mg in the soil could be detrimental to plant growth and therefore plants have evolved multiple adaptive mechanisms to maintain cellular Mg concentration within an optimal range [30]. In higher plants, the most well-documented Mg^2+^ transporters (MGTs) belong to homologues of bacterial CorA superfamily and are also called “MRS2” based on their similarity to yeast Mitocondrial RNA splicing 2 protein [31,32]. Several members of the MGT family mediate Mg^2+^ transport in bacteria or yeast as indicated by functional complementation as well as ^63^Ni tracer assay [31,32,33]. In plants, they have been shown to play vital roles in Mg^2+^ uptake, translocation, and homeostasis associated with their different subcellular localizations and diverse tissue-specific expression patterns [30]. For instance, MGT2 and MGT3 are tonoplast localized and possibly involved in Mg^2+^ partitioning into mesophyll vacuoles [34]; MGT4, MGT5, and MGT9 are strongly expressed in mature anthers and play a crucial role in pollen development and male fertility [35,36,37,38]. MGT6 and MGT7 are shown to be most directly involved in Mg homeostasis because knocking-down or knocking-out either of the genes leads to hypersensitivity to low Mg conditions [33,39]. MGT6 encodes a plasma membrane-localized high-affinity Mg^2+^ transporter and mediates Mg^2+^ uptake in root hairs, particularly under Mg-limited conditions [39]. MGT7 is also preferentially expressed in roots and loss-of-function of MGT7 caused poor seed germination and severe growth retardation under low-Mg conditions [33]. Double mutant of *mgt6* and *mgt7* displayed a stronger phenotype than single mutants, suggesting that MGT6 and MGT7 may be synergistic in controlling Mg homeostasis in low-Mg environment conditions [40].

In contrast to considerable research on Mg transport and homeostasis under Mg deficient conditions, the regulatory mechanisms required for adaptation to excessive external Mg remain poorly understood. Recent studies suggested that MGT6 and MGT7 are essential for plants to adapt to both normal and high Mg conditions [40,41]. The *mgt6* mutant displayed dramatic growth defects with a decrease in cellular Mg content in the shoot, when grown under high Mg^2+^. Grafting experiments further suggested a shoot-based mechanism for Mg^2+^ detoxification although the exact role of MGT6 in this process is still not clear. More importantly, a core regulatory pathway consisting of two calcineurin B-like Ca sensors (CBL2 and CBL3) partnering with four CBL-interacting protein kinases (CIPK; CIPK3/9/23/26) has been established that allows plant cells to sequester Mg^2+^ into plant vacuoles, thereby protecting plant cells from high Mg^2+^ toxicity [42]. In this study, we identified the tonoplast pyrophosphatase, AVP1, as an important component in high Mg^2+^ tolerance in *Arabidopsis*. Furthermore, by analyzing the *avp1-4 mgt6* double mutant and *avp1-4 cbl2 cbl3* triple mutant, we showed that the role of AVP1 in high-Mg tolerance was independent of previously reported MGT6 or CBL/CIPK-mediated pathway. Instead, our results suggested a novel link between high Mg^2+^ stress and PPi homeostasis in plants.

## 2. Results

### 2.1. The avp1 Mutant Is Hypersensitive to High External Magnesium Conditions

The originally reported T-DNA insertional mutant *avp1-1* contains an additional T-DNA insertion causing phenotypes unrelated to AVP1 mutation [22,43]. We thus characterized another T-DNA insertion line *avp1-4* (GK-596F06) for this study. The *avp1-4* mutant carried a T-DNA insertion in the third exon of *AVP1* as further confirmed by PCR analysis and DNA sequencing (Figure 1a). The *avp1-4* homozygous mutants lacked detectable *AVP1* transcripts (Appendix A), and its tonoplast PPi hydrolysis activity was considerably diminished, to only 10% of wild type (Appendix A). Compared with wild-type plants (Col-0), *avp1-4* mutants exhibited no obvious phenotypic changes during the life cycle including vegetative and reproductive periods (Appendix A), which is quite different from *avp1-1* [43], because pleiotropic phenotypes observed in *avp1-1* are caused by mutation in the *GNOM* (At1g13980) gene [22]. We examined the phenotype of *avp1-4* plants under multiple ionic stress conditions and found that *avp1-4* mutant and wild-type seedlings grew similarly on the MS medium and did not show hypersensitive response to most of the ionic stresses such as 60 mM Na^+^, 60 mM K^+^, 40 mM Ca^2+^, 100 µM Zn^2+^, 40 µM Cu^2+^, or 100 µM Fe^3+^ (Appendix A). However, the growth of *avp1-4* seedlings were severely impaired when 20 mM MgCl_2_ was Appendix A. To validate the hypersensitivity of *avp1-4* to MgCl_2_, we grew the seedlings of the mutant together with the wild-type plants on the 1/6 MS medium containing various levels of Mg^2+^, the *avp1-4* mutant plants were clearly stunted as compared with Col-0 (Figure 1b), although the primary root length of *avp1* was comparable to that of Col-0 (Figure 1d). In addition, we also studied one more mutant allele of *AVP1* gene in the Wassilewskija (Ws) background, designated as *avp1-3*, and another three mutant alleles of AVP1, *fugu5-1*, *fugu5-2*, and *fugu5-3* in the Col-0 background [13] (Figure 1a). Measurements of seedling fresh weight confirmed a severe growth inhibition by 8 mM MgCl_2_ in both *avp1-4* and *avp1-3* mutants, as compared with their respective wild-type counterparts (Figure 1e). Consistently, we also found that high-Mg sensitivity phenotypes in the three *fugu5* mutants were comparable to those in *avp1-4* (Figure 1c). Together, these results suggested that AVP1 is required for Mg^2+^ tolerance in *Arabidopsis*.

### 2.2. The Enzymatic Pyrophosphatase Activity Is Required for High-Mg Tolerance in Plants

To verify that the observed phenotypes in the *avp1* mutants are caused by a defect in AVP1, we conducted a complementation test in *avp1-4* background. A coding sequence fragment of AVP1 was introduced into the *avp1-4* mutant, and several homozygous transgenic lines were obtained (Appendix A). Phenotypic analysis of two representative lines showed that oblong-shaped cotyledons of *avp1-4* when germinated on MS media containing low sucrose or in soil were fully restored to normal shape (Appendix A). In addition, seedling growth defects of *avp1-4* under high-Mg conditions were also completely rescued (Figure 2a). Root length and shoot fresh weight of the transgenic lines under high Mg conditions were similar to those of the wild type (Figure 2b,c). These data further confirmed that loss-of-function in AVP1 was indeed the causal mutation for the high-Mg hypersensitive phenotype of *avp1-4*.

Reducing the PPi concentration in the cytoplasm and increasing the acidification of vacuoles represent the two main biochemical functions of AVP1. In order to dissect if both activities are required in this specific high Mg^2+^-associated process, we resorted to the transgenic line expressing yeast *IPP1* gene under the control of the *AVP1* promoter in the *fugu5-1* mutant background [13]. IPP1 is a cytosolic soluble protein which is not capable of translocating H^+^, thus decoupling the hydrolysis and proton pump activities. Interestingly, our results showed that the severely retarded growth of *fugu5-1* mutant plants under high-Mg conditions was completely recovered by expression of the *IPP1* gene (Figure 2d). The quantitative analysis of seedling fresh weight confirmed the complementation (Figure 2e,f).

To extend the phenotypic analysis of the *avp1* mutants in mature plants, we examined the phenotype of *avp1* mutants using hydroponic culture system. Consistent with the patterns of plant growth on agar plates, the mutant plants exhibited a pronounced growth defect (Figure 2g) than wild-type plants in the hydroponic solutions supplemented with 15 mM external Mg^2+^, as revealed by much lower fresh weight (Figure 2h) and lower chlorophyll content (Figure 2i). The *IPP1* transgenic line also behaved like wild-type plants but not *avp1* mutant under this condition, suggesting that PPi hydrolysis is the key function that AVP1 plays in high-Mg adaptation.

To address the contribution of PPi hydrolysis activity to high-Mg tolerance, we directly measured V-PPase activity and PPi content under normal and high-Mg conditions. Under normal conditions, PPi hydrolysis activity of two *avp1* mutant alleles was reduced by ∼85%, whereas activity from two complementary lines was comparable to the wild-type control (Figure 3a). Consistently, the amount of PPi from both mutants was increased by ∼50% (Figure 3b). After grown for three days on 15 mM Mg^2+^, all the plants displayed reduced PPi hydrolysis activity and higher PPi content. However, the PPi elevation of mutant plants during high Mg^2+^ stress was significantly higher than that of wild type (Figure 3). Altogether, these results strongly indicate that the dampened hydrolysis of cytosolic PPi is the major reason for the increased Mg sensitivity in the *avp1* mutants.

### 2.3. The avp1 Mutant Is Not Compromised in Mg^2+^ Homeostasis

To assess whether increased Mg^2+^ sensitivity in the *avp1* mutant is associated with Mg^2+^ homeostasis, we measured the Mg content in wild-type (Col-0 and Ws) and mutant plants (*avp1-4* and *avp1-3*) using ICP-MS. When 8 mM Mg^2+^ was added to the growth medium, Mg content in either shoot or root in all the plants was strikingly elevated, but no significant difference between wild-type and mutant plants in Mg content was observed. (Figure 4a,b). Considering Ca and Mg often affect each other in their uptake and transport [30], we also measured the Ca content in the same plants. Consistent with Mg-Ca antagonism, the Ca content in both wild-type and *avp1* mutant plants was evidently lower when plants were grown under high external Mg^2+^ conditions, but Ca content in the shoots and roots in *avp1* mutants was similar to that in wild-type plants (Figure 4c,d). These data suggest that both Mg and Ca homeostasis are not altered in the *avp1* mutants, which are consistent with the earlier conclusion that PPi hydrolysis rather than vacuolar acidification is responsible for AVP1 function under high-Mg stress.

### 2.4. AVP1 and MGT6 Function Independently in High-Mg Tolerance in Arabidopsis

In *Arabidopsis*, the magnesium transporter MGT6 is important for controlling plant Mg^2+^ homeostasis and adaptation to both low- and high-Mg conditions [39,40,41]. To investigate the functional interaction between AVP1 and MGT6, we created a double mutant that lacks both *AVP1* and *MGT6* transcripts (Figure 5a). We next tested the sensitivity of *avp1-4 mgt6* double mutant to high external Mg conditions. When grown on the 1/6 MS medium containing 0.25 mM Mg^2+^, the *mgt6* and *avp1-4 mgt6* plants showed obvious growth retardation compared with Col-0 and *avp1-4* seedlings, resulting from *mgt6* mutation that renders plants hypersensitive to low Mg^2+^ (Figure 5b,f). When the medium Mg^2+^ levels reached 1 mM, the growth of *mgt6* and *avp1-4 mgt6* mutants appeared comparable to that of wild-type (Figure 5c,f). Notably, in the presence of high Mg levels such as 4 mM and 6 mM Mg^2+^, *avp1-4 mgt6* double mutant exhibited more severe inhibition of shoot growth with significantly lower fresh weight (Figure 5d–f) and more reduced chlorophyll content (Figure 5g) as compared to either *mgt6* or *avp1* single mutant. The enhanced sensitivity of the *avp1-4 mgt6* double mutant suggest that AVP1 and MGT6 may represent two independent functions that are required for plant tolerance to high Mg^2+^ stresses.

### 2.5. Hypersensitivity of avp1-4 cbl2 cbl3 Triple Mutant to High External Mg^2+^ Concentrations

The vacuolar Mg^2+^ sequestration regulated by tonoplast-localized CBL-CIPK modules is established as a key mechanism in detoxifying excessive Mg^2+^ in plant cells [42]. Since AVP1 also resides in the tonoplast [44], it is relevant to examine whether the hypersensitivity of *avp1* mutants would be somehow associated with the vacuolar Mg^2+^ sequestration controlled by CBL2 and CBL3. To this end, we constructed an *avp1-4 cbl2 cbl3* triple mutant (Figure 6a) and subsequently characterized its phenotype in the presence of various Mg^2+^ concentrations (Figure 6b–e). The wild type and the three mutants, *avp1-4*, *cbl2 cbl3,* and *avp1-4 cbl2 cbl3,* grow normally on 1/6 MS containing 0.25 mM or 1 mM Mg^2+^. Under high Mg^2+^ conditions (4 mM, 6 mM), both *avp1-4* and *cbl2 cbl3* mutants displayed severe growth retardation, whereas *avp1-4 cbl2 cbl3* triple mutant hardly survived, displaying an even more sensitive phenotype to high Mg^2+^ than either of the single mutants (Figure 6d,e). Measurements of seedling fresh weight and leaf chlorophyll content confirmed much more severe growth inhibition by 4 mM and 6 mM MgCl_2_ in the *avp1-4 cbl2 cbl3* triple mutant (Figure 6f,g). We concluded that the Mg hypersensitivity of *avp1* mutants results from altered processes independent of vacuolar Mg^2+^ partitioning pathway regulated by CBL2 and CBL3.

## 3. Discussion

Although Mg is an essential macronutrient required for plant growth, high concentrations of environmental Mg^2+^ could be detrimental, and the targets underlying toxic effect of high-Mg are not well understood. In the present study, we characterized multiple *avp1* mutant alleles and found they were hypersensitive to high external Mg^2+^. This finding has not only improved our understanding of the mechanism underlying Mg^2+^ tolerance but also uncovered a novel physiological function of AVP1 in plants. When the plants were confronted with high Mg stress, sequestration of excessive Mg^2+^ into the vacuole plays a vital role in detoxification of Mg excess from the cytoplasm [30,45]. The AVP1 protein predominantly localized in the vacuolar membrane [44] and was a highly abundant component of the tonoplast proteome [21]. Encoded by AVP1, vacuolar H^+^-PPase, together with vacuolar H^+^-ATPase, plays a critical part in establishing the electrochemical potential by pumping H^+^ across the vacuolar membrane. This proton gradient, in turn, facilitates secondary fluxes of ions and molecules across the tonoplast [21,22,27]. Based on this well-established idea, we hypothesized that *avp1* mutants may be impaired in cellular ionic homeostasis and should thus exhibit hypersensitivity to a broad range of ions. However, unexpectedly, we found that *avp1* was hypersensitive only to high external Mg^2+^ but not to other cations (Appendix A). It was shown that overexpression of AVP1 improved plant salt tolerance in quite a few species, which was interpreted as the result of increased sequestration of Na^+^ into the vacuole [23,46,47]. It is thus reasonable to speculate that the tonoplast electrochemical potential generated by AVP1 would likewise favor Mg^2+^ transport into vacuoles via secondary Mg^2+^/H^+^ antiporter. Surprisingly, our subsequent experiments did not support this hypothesis and several lines of evidence suggested that the hypersensitivity of *avp1* to high Mg^2+^ was not due to the compromised Mg^2+^ homeostasis in the mutant. First, unlike other high Mg^2+^-sensitive mutants such as *mgt6* and the vacuolar *cbl/cipk* mutants, the Mg and Ca content in the *avp1* mutant was not altered as compared with wild type, suggesting that AVP1 may not be directly involved in Mg^2+^ transport in plant cells. Second, higher order mutants of the *avp1-4 mgt6* double mutant and *avp1-4 cbl2 cbl3* triple mutant displayed a dramatic enhancement in Mg^2+^ sensitivity as compared to single mutants. These genetic data strongly suggest that AVP1 does not function in the same pathway mediated by MGT6 and does not serve as a target for vacuolar CBL-CIPK. Moreover, it was previously shown that either vacuolar H^+^-ATPase double mutant *vha-a2 vha-a3* or the *mhx1* mutant defective in the proposed Mg^2+^/H^+^ antiporter was not hypersensitive to high Mg^2+^ [42]. These results implicate the vacuolar Mg^2+^ compartmentalization should be fulfilled by an unknown Mg^2+^ transporter/channel, whose activity is largely not dependent on the tonoplast ΔpH. Identification of this novel Mg^2+^ transport system across the tonoplast, which is probably targeted by vacuolar CBL-CIPK complexes, would be the key to understand the mechanism. Third, expression of the cytosolic soluble pyrophosphatase isoform *IPP1* could fully rescue the Mg-hypersensitivity caused by *AVP1* mutation. These lines of evidence pinpoint PPi hydrolysis, rather than ΔpH-assisted secondary ion transport and sequestration, as the major function of AVP1 in high Mg^2+^ adaptation.

Under high Mg stress conditions, a number of adaptive responses are supposed to take place in plants, including the remodeling of plant morphogenesis as well as reprogramming of the gene expression and metabolite profile. However, very little is known so far and therefore, the molecular components targeted by excessive Mg^2+^ in plant cells remain obscure. Here, we suggest that the concentration of cellular PPi could be responsive to external Mg supply. Our results showed that extremely high levels of Mg^2+^ led to inhibition of the PPase activity in *Arabidopsis*, which in turn, resulted in the elevation of PPi content in the cytosol. Because high level of PPi is very toxic, the efficient removal of PPi by AVP1 under high Mg^2+^ conditions might become one of the limiting factors for optimal plant growth. This idea is supported by the observation that *avp1* mutants accumulated significantly higher PPi content under high Mg^2+^ conditions compared with normal conditions (Figure 3). Most importantly, heterologous expression of the soluble PPase *IPP1* gene rescued high Mg-sensitive phenotype of *fugu5-1* (Figure 2), which strongly suggested that high Mg^2+^ hypersensitivity phenotype in *avp1* mutants could primarily be attributed to impaired PPi homeostasis. It would be interesting to investigate how PPi concentrations vary in different Mg^2+^ conditions and during different plant growth stages. Recently, cytosolic soluble pyrophosphatases (AtPPa1 to AtPPa5) were identified in *Arabidopsis*, and were shown to physiologically cooperate with the vacuolar H^+^-PPase in regulating cytosolic PPi levels [16]. Future studies should clarify if this type of soluble isoenzymes is also involved in the same high-Mg adaptation process. Collectively, our findings provide genetic and physiological evidence that *AVP1* is a new component required for plant growth under high external Mg^2+^ concentrations and functions in regulating Mg^2+^ tolerance via PPi hydrolysis.

## 4. Materials and Methods

### 4.1. Plant Materials and Growth Conditions

*Arabidopsis thaliana* ecotype Columbia (Col-0) and Wassilewskija (Ws) were used as wild type in this study. The mutants *fugu5-1*, *fugu5-2*, *fugu5-3*, and transgenic plants *fugu5-1*+*IPP1* were offered and characterize by Ferjani (2011) [13]. The *cbl2 cbl3* double mutant was described in previous studies [48]. The T-DNA insertion mutants *avp1-4* (GK-596F06) and *mgt6* (SALK_205483) were obtained from the European Arabidopsis Stock Centre and the Arabidopsis Biological Resource Center. The mutant *avp1-3* (FLAG_291B12) was a T-DNA insertion mutant in the Wassilewskija (Ws) background and obtained from INRA Arabidopsis T-DNA mutant library. Mutants with multiple gene-knockout events were generated by genetic crosses, and homozygous mutant plants were screened from F2 generation and identified by genomic PCR using primers listed in Appendix A.

### 4.2. Phenotypic Analysis

For on-plate growth assays, seeds of different genotypes were sterilized with 75% ethanol for 10 min, washed in sterilized water for three times, and sown on Murashige and Skoog (MS) medium containing 2% sucrose (Sigma) and solidified with 0.8% phytoblend (Caisson Labs). The plates were incubated at 4 °C in darkness for two days and then were positioned vertically at 22 °C in growth chamber with a 14 h light/10 h dark photoperiod. After germination, five-day-old seedlings were transferred onto agarose-solidified media containing various ions as indicated in the figure legends and were grown under 14 h light/10 h dark photoperiod.

For phenotypic assay in the hydroponics, 10-day-old seedlings geminated on MS plate were transferred to 1/6 strength MS solution and were grown under the 14 h light/10 h dark condition in the plant growth chamber. Fresh liquid solutions were replaced once a week. After two-week culture, the plants were treated with 1/6 MS solutions supplemented with 15 mM MgCl_2_.

### 4.3. Crude Membrane Preparation and Enzymatic Activity Assays

Two-week-old hydroponically grown plants were treated with 1/6 MS solutions containing 0 or 15 mM MgCl_2_. After two-day treatment, leaves of all the plants were collected to prepare crude membrane as described previously [48]. Plant materials were ground at 4 °C with cold homogenization buffer containing 350 mM sucrose, 70 mM Tris-HCl (pH 8.0), 3 mM Na_2_EDTA, 0.2% (*w*/*v*) BSA, 1.5% (*w*/*v*) PVP-40, 5 mM DTT, 10% (*v*/*v*) glycerol, 1 mM PMSF and 1×protease inhibitor mixture (Roche). The homogenate was filtered through four layers of cheesecloth and centrifuged at 4000× *g* for 20 min at 4 °C. The supernatant was then centrifuged at 100,000× *g* for 1 h. The obtained pellet was suspended in 350 mM sucrose, 10 mM Tris-Mes (pH 7.0), 2 mM DTT and 1× protease inhibitor mixture.

Pyrophosphate hydrolysis was measured as described in previous studies [48]. The assay solution for PPi hydrolysis activity contained 25 mM Tris-Mes (pH 7.5), 2mM MgSO_4_, 100 µM Na_2_MoO_4_, 0.1% Brij 58, and 200 µM Na_4_P_2_O_7_. PPase activity was expressed as the difference of phosphate (Pi) release measured in the absence and the presence of 50 mM KCl. After incubation at 28 °C for 40 min, 40 mM citric acid was added to terminate reactions. For the measurement of inorganic Pi amount, freshly prepared AAM solution (50% (*v*/*v*) acetone, 2.5 mM ammoniummolybdate, 1.25 M H_2_SO_4_) was added to the reaction solution, vortexed and colorimetrically examined at 355 nm.

### 4.4. Quantification of Pyrophosphate in Plants

Two-week-old hydroponically grown plants were transferred to 1/6 MS solutions containing 0 or 15 mM MgCl_2_. After two-day treatment, leaves of all the plants were collected and PPi was extracted from leaf tissue as described previously [49]. Leaf samples were ground to powder in liquid nitrogen, suspended with three volumes of pure water, heated at 85 °C for 15 min, and then centrifuged at 15,000 rpm for 10 min. The supernatants were collected and then centrifuged at 40,000 rpm for 10 min. The obtained supernatants were diluted with pure water and subjected to PPi assay using a PPi Assay Kit (Sigma, St. Louis, MO, USA) according to the manufacturer’s instructions. Fluorescence was monitored with a Safire 2 plate reader set at 316 nm for excitation and 456 nm for emission (Tecan, Männedorf, Switzerland).

### 4.5. Measurements of Mg and Ca Content

One-week-old Arabidopsis seedlings were transferred onto 1/6-strength MS medium supplemented with 0 or 8 mM MgCl_2_. After a seven-day treatment, seedlings of wild-type and mutant plants were collected and pooled into roots and shoots. The samples were washed with 18 MΩ water for three to five times, dried for 48 h at 80 °C, milled to fine powder, weighed, and digested with concentrated HNO_3_ (Sigma-Aldrich, Milwaukee, WI, USA) in 100°C water bath for 1 h. Mg^2+^ and Ca^2+^ concentrations were determined using an ICP mass spectrometer (PerkinElmer NexION 300). Each sample was tested three times.

### 4.6. Statistical Analysis of the Data

All data in this work were obtained from at least three independent experiments. Data were subjected to statistical analyses using Student’s *t*-test (*p* < 0.05) or one-way analysis of variance (ANOVA) followed by Duncan’s multiple range test (*p* < 0.05).

## Figures and Tables

**Figure 1 ijms-19-03617-f001:**
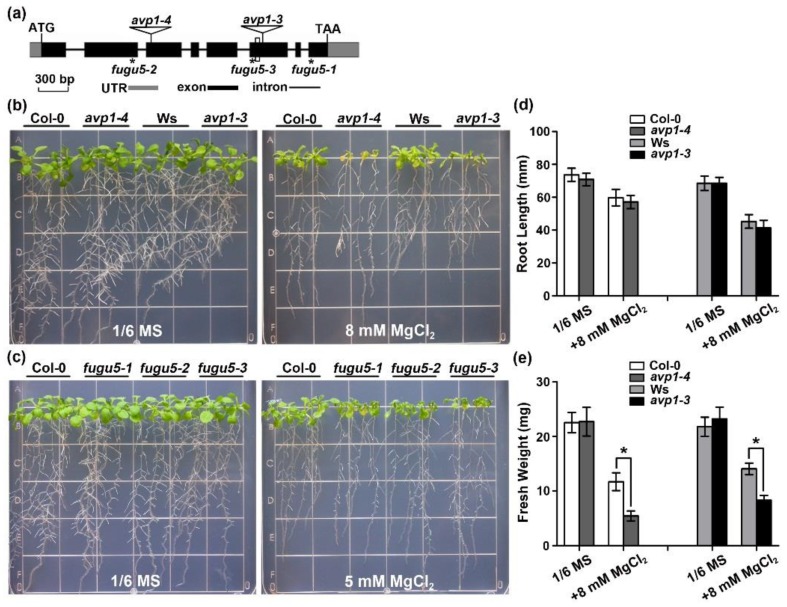
The *avp1* mutant plants are hypersensitive to high-Mg conditions. (**a**) Schematic diagram of the *AVP1* gene. Coding regions are depicted as black boxes, and the 5’ or 3’ UTRs are shown as shaded boxes and lines represent introns to scale. The molecular lesion in each of the five loss-of-function *avp1* alleles is indicated by open triangles or asterisks. (**b**) Growth phenotype of wild types Col-0 and Ws and corresponding mutant plants *avp1-4* and *avp1-3* under different ionic stress conditions. Five-day-old seedlings were transferred onto 1/6 MS medium or 1/6 MS medium supplemented with 8 mM MgCl_2_. Photographs were taken on the 10th day after transfer. (**c**) Growth phenotype of Col-0 and three *fugu5* mutant alleles on 1/6 MS medium or 1/6 MS medium supplemented with 5 mM MgCl_2_. (**d**) Root length and (**e**) fresh weight of seedlings on the 10th day after transfer. Data are presented as the mean ± SD of four replicate experiments. Asterisks indicate statistically significant differences compared with the Col-0 or Ws (Student’s *t*-test, * *p* < 0.05).

**Figure 2 ijms-19-03617-f002:**
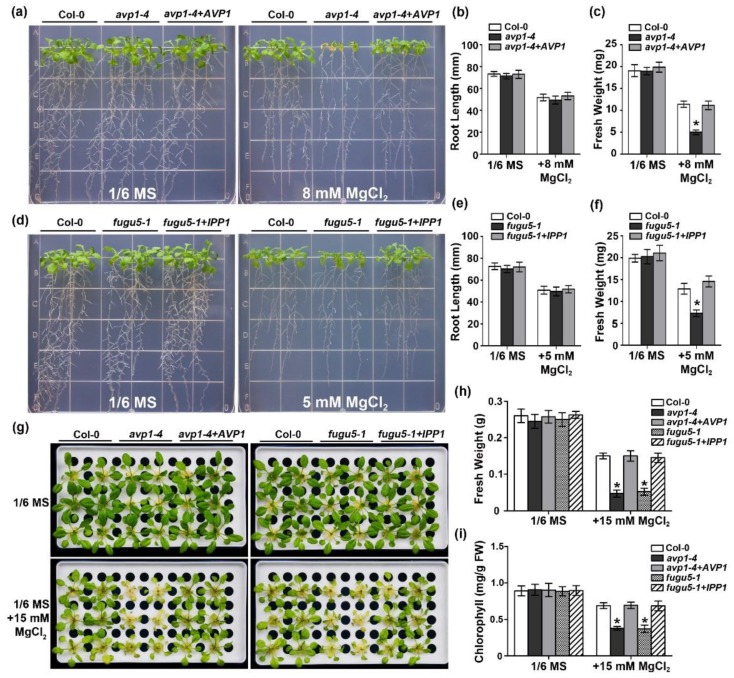
Functional complementation of *avp1-4* and *fugu5-1* under high-Mg stress grown on agar plates or in hydroponic culture. (**a**) Growth phenotype of wild type Col-0, *avp1-4* and corresponding complementary line *avp1-4*+*AVP1* on 1/6 MS medium or 1/6 MS medium supplemented with 8 mM MgCl_2_. Photographs were taken on the 10th day after transfer. (**b**) Root length and (**c**) fresh weight of seedlings described in (a) on the 10th day after transfer. (**d**) Growth phenotype of wild type (Col-0), *fugu5-1* and corresponding complementary line *fugu5-1*+*IPP1* on 1/6 MS medium or 1/6 MS medium supplemented with 5 mM MgCl_2_. Photographs were taken on the 10th day after transfer. (**e**) Root length and (**f**) fresh weight of seedlings described in (d) on the 10th day after transfer. (**g**) Effect of Mg^2+^ concentration on growth of Col-0, *avp1-4*, *fugu5-1*, and corresponding complemented lines. Two-week-old plants grown in 1/6 MS hydroponic culture were transferred onto 1/6 MS or 1/6 MS hydroponic culture supplemented with 15 mM MgCl_2_. Photographs were taken on the 5th day after transfer. (**h**) Fresh weight and (**i**) chlorophyll content of the plants of various genotypes described in (**g**). Data are presented as the mean ± SD of four replicate experiments. Asterisks indicate statistically significant differences compared with the Col-0 (Student’s *t*-test, * *p* < 0.05).

**Figure 3 ijms-19-03617-f003:**
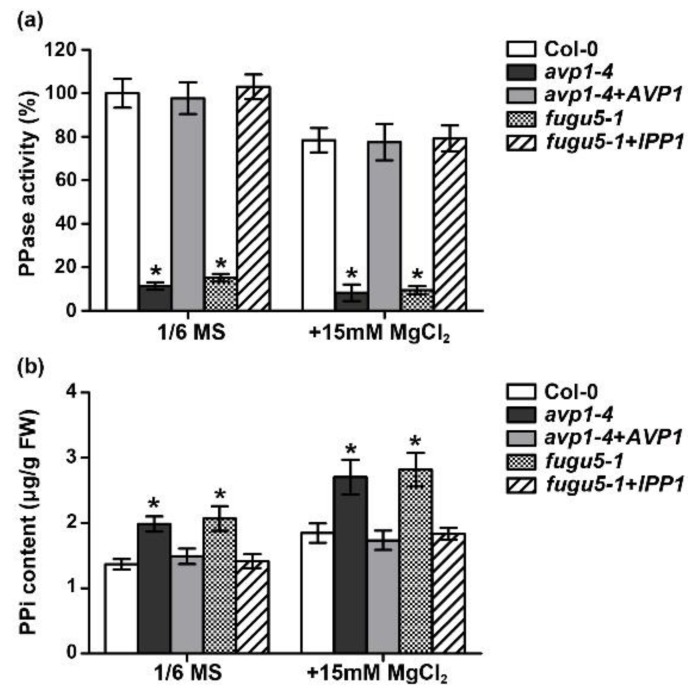
V-PPase activities and the contents of PPi under different Mg^2+^ conditions. (**a**) H^+^-PPase hydrolysis activity was determined from microsomal membranes of two-week old plants of wild type (Col-0), mutants (*avp1-4* and *fugu5-1*) and complementary lines (*avp1-4*+*AVP1* and *fugu5-1*+*IPP1*) grown in hydroponic culture. Results are shown as percentage of the Col-0 control activity. Values are mean ± SD of three replicate experiments. (**b**) PPi content of seedlings under different Mg^2+^ conditions. Seedlings grown under the same conditions and collected at the same stage as described in (**a**) were used for the quantification of PPi. Values are mean ± SD from triplicate experiments. Asterisk indicates significant difference compared with the wild type (Student’s *t*-test, * *p* < 0.05).

**Figure 4 ijms-19-03617-f004:**
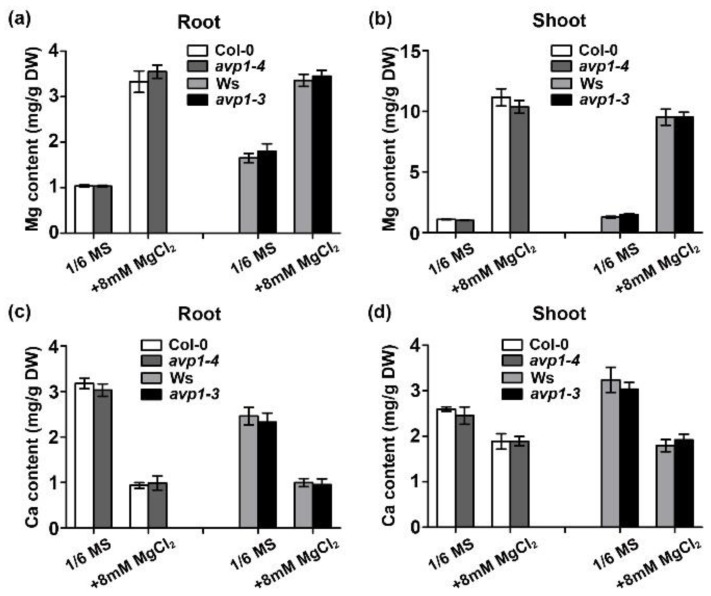
Mg and Ca content in the *avp1* mutant under different Mg^2+^ conditions. (**a**,**b**) Mg content in the root (a) and shoot (b) under different Mg^2+^ regimes. (**c**,**d**) Ca content in the root (c) and shoot (d) under different Mg^2+^ regimes. Data are presented as the mean ± SD of triplicate experiments.

**Figure 5 ijms-19-03617-f005:**
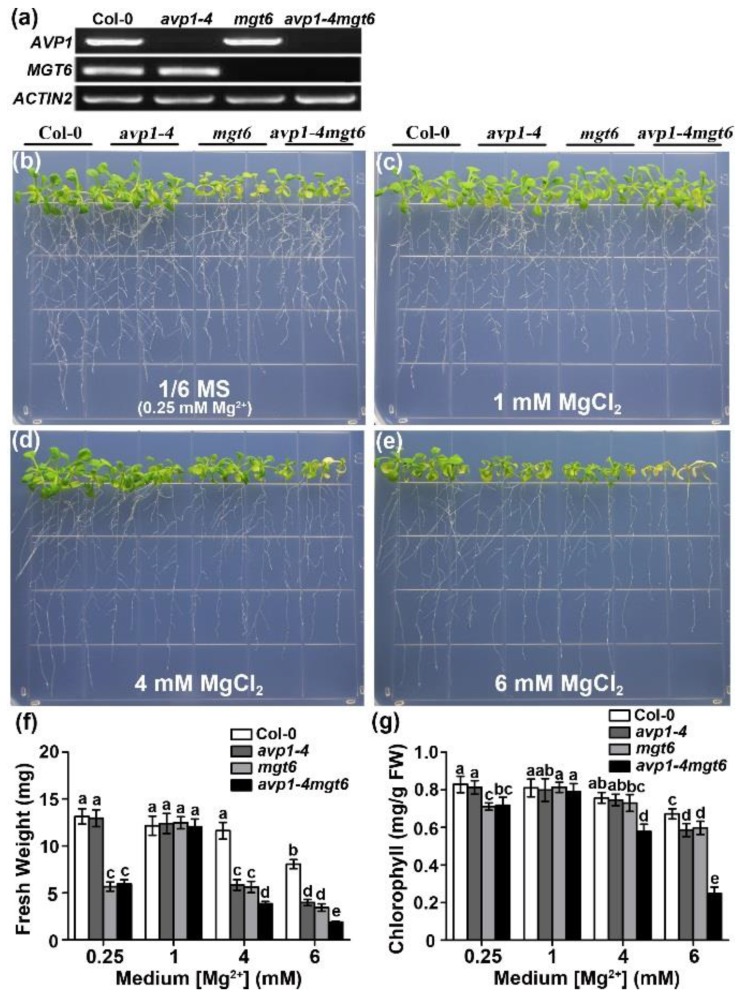
Phenotypic analysis of Mg^2+^ sensitivity in the *avp1-4 mgt6* double mutant. (**a**) RT-PCR analysis of *AVP1* and *MGT6* gene expression in wild-type Col-0, homozygous *avp1-4* or *mgt6* single mutant and the *avp1-4 mgt6* double mutant. (**b**–**e**) Phenotypic analysis of Mg^2+^ sensitivity in *avp1-4*, *mgt6*, and *avp1-4 mgt6* mutants. (**f**) Fresh weight of seedlings on the 10th day after transfer. (**g**) Chlorophyll content of seedlings on the 10th day after transfer. Data are mean ± SD from triplicate experiments. Any pair of genotypes/treatments that do not share the same letter are significantly different (*p* < 0.05) based on a Duncan’s multiple range test.

**Figure 6 ijms-19-03617-f006:**
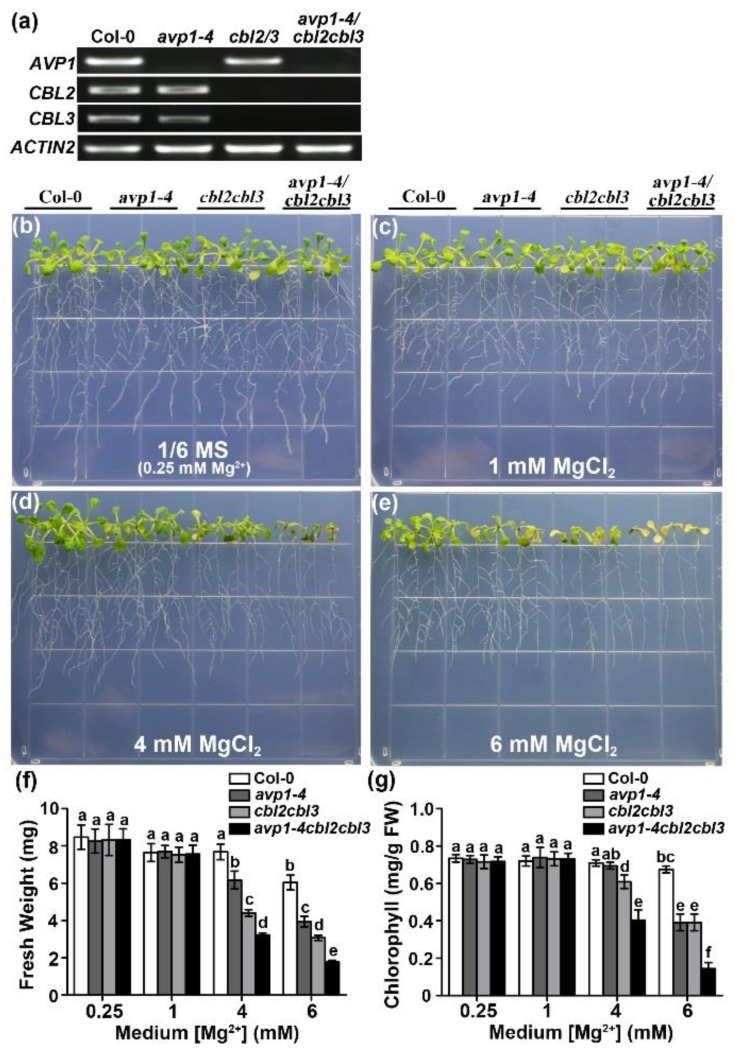
Phenotypic analysis of Mg^2+^ sensitivity in the *avp1-4 cbl2 cbl3* triple mutant. (**a**) RT-PCR analysis of *AVP1*, *CBL2*, and *CBL3* gene expression in wild type Col-0, *avp1-4* single mutant, *cbl2 cbl3* double mutant and *avp1-4 cbl2 cbl3* triple mutant. (**b**–**e**) Phenotypic analysis of Mg^2+^ sensitivity in *avp1-4*, *cbl2 cbl3*, and *avp1-4 cbl2 cbl3* mutants. (**f**) Fresh weight of seedlings on the 10th day after transfer. (**g**) Chlorophyll content of seedlings on the 10th day after transfer. Data are mean ± SD from triplicate experiments. Any pair of genotypes/treatments that do not share the same letter are significantly different (*p* < 0.05) based on a Duncan’s multiple range test.

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
