# Peer review of "Vacuolar Proton Pyrophosphatase Is Required for High Magnesium Tolerance in Arabidopsis"

_ijms, 2018, doi:10.3390/ijms19113617_

Round 1
Reviewer 1 Report
Yang and colleagues report on the essential function of vacuolar proton pyrophosphatase for high magnesium tolerance in Arabidopsis. The mechanism underlying magnesium tolerance remains to be elucidated. The authors claimed that high magnesium caused excess PPi contents in avp1 mutants, which caused growth defects. The growth defect was rescued by IPP1, which hydrolyze PPi. Their presentation and discussion seems to be relevant and provides the readers new mechanism of high magnesium tolerance in Arabidopsis. I recommend the article for publication.
Author Response
Dear reviewer,
We are grateful for your approval and constructive comments which helped us improve our manuscript during this revision. We made two major changes according to another reviewer’s suggestions, which have been highlighted in green in the text. First, we re-designated the mutant line “GK-596F06” as “avp1-4” (which was “avp1-2” potentially causing confusion due to contradiction to old literature). Second, some research background with regard to the PPi synthesis activity of AVP1 was added in the Introduction and more references were included accordingly.
Thank you for your consideration.
Reviewer 2 Report
In the present manuscript authors studied the mechanism underlying Mg2+ tolerance in plants (Arabidopsis). The main conclusion of this work “AVP1 is required for cellular PPi homeostasis that in turn contributes to high-Mg tolerance in plant cells” is well supported with experimental data (genetic and physiological evidences), and results are solid. Authors uncovered a novel physiological function of AVP1 in plants, related with Mg toxicity tolerance. The manuscript is well written, understandable, and appropriate for the scope of “International Journal of Molecular Science”.
Minor points
Introduction:
Given the importance of PPi homeostasis in high-Mg tolerance revealed in this work, there must be a mention or discussion about previous works that showed the PPi synthesis activity of plant Proton-Pumping Pyrophosphatases (Rocha Facanha and de Meis 1998; Marsh et. al., 2000; Gaxiola et al., 2012; Pizzio et al., 2015; Khadilkar et. al., 2016).
Results:
· The avp1-2 line was previously defined as the T-DNA promoter insertion line SALK-046492 (Pizzio et. al., 2015). To avoid future confusions, I suggest to rename the line used in the present manuscript GK-596F06 (e.g. avp1-4).
· Figure 1a: The mutant avp1-3 representation hints that is a T-DNA insertion line, and it is not. Symbol should be changed.
· Figure S1b upper panel: The PCR with For/LB primers should be negative in avp1-2 line, Is For/RB the right primer pair?
Materials and Methods:
· Give more information about the line avp1-3. What kind of mutation does avp1-3 line carry?
Author Response
Point-by-point response to Reviewer 2’s Comments
1: Given the importance of PPi homeostasis in high-Mg tolerance revealed in this work, there must be a mention or discussion about previous works that showed the PPi synthesis activity of plant Proton-Pumping Pyrophosphatases (Rocha Facanha and de Meis 1998; Marsh et. al., 2000; Gaxiola et al., 2012; Pizzio et al., 2015; Khadilkar et. al., 2016).
Response: This is a constructive suggestion, which certainly help to reinforce the core idea of the work. As suggested, more background was included in both the introduction and discussion on PPi synthesis activity of AVP1, along with more references (green highlighted in the text).
2: The avp1-2 line was previously defined as the T-DNA promoter insertion line SALK-046492 (Pizzio et. al., 2015). To avoid future confusions, I suggest to rename the line used in the present manuscript GK-596F06 (e.g. avp1-4).
Response: As suggested, the line GK-596F06 was renamed as “avp1-4” throughout the revised manuscript.
Point 3: Figure 1a: The mutant avp1-3 representation hints that is a T-DNA insertion line, and it is not. Symbol should be changed.
Response: After we double checked the original data, in particular with the author who was responsible for identification of avp1-3, we confirmed that it is actually a T-DNA insertion mutant in the sixth exon. We have corrected the description on that allele.
Point 4: Figure S1b upper panel: The PCR with For/LB primers should be negative in avp1-2 line, Is For/RB the right primer pair?
Response: Yes, For/LB is the right primer pair. The primers “For” and “Rev” amplifies the sequence from the AVP1 gene, and the primer “LB” was designed based on the “left boarder” sequence from T-DNA in vector pAC106, which was carried in GABI-Kat mutants. Therefore, For/LB and Rev/LB primer pairs were used to detect whether there is any T-DNA insertion in the genomic region of AVP1. Because a double-inverted T-DNA insertion was found in that site (which is quite common in many T-DNA insertional mutants), the PCR amplification results using both For/LB and Rev/LB primer combinations are positive in avp1-2.
Point 5: Give more information about the line avp1-3. What kind of mutation does avp1-3 line carry?
Response: It is actually another T-DNA insertion mutant from INRA Arabidopsis T-DNA mutant library. Those T-DNA insertion lines were built in the Ws (Wassilewskija) ecotype background and the ID of avp1-3 was “FLAG_291B12”. Detailed information of this allele was added in the method section of the revised manuscript.